# Bentonite clay with different nitrogen sources can effectively reduce nitrate leaching from sandy soil

Zahid Hussain[1¤]*, Tang Cheng[1]*, Muhammad Irshad[2], Riaz Ahmed Khattak[3], Chen Yao[4], Di Song[1], Muhammad Mohiuddin[5]

**1** Yunnan Key Laboratory of Pollution Process and Management of Plateau-Lake -Watershed, Yunnan Research Academy of Eco-Environmental Sciences, Kunming, China, **2** Department of Environmental Sciences, CUI, Abbottabad Campus, Abbottabad, Pakistan, **3** The Brains Institute, Peshawar, Pakistan, **4** Yunnan Infrastructure Investment Co. Ltd., Kunming, China, **5** Department of Environmental Sciences, Kohsar University, Murree, Pakistan

¤ Current address: Department of Development Studies, COMSATS University Islamabad (CUI), Abbottabad Campus, Abbottabad, Pakistan
* drzahid@cuiatd.edu.pk (ZH); 1290992119@qq.com (TC)

**Data Availability Statement:** All relevant data are within the paper and its Supporting information files.

## Abstract

Nitrate ($NO_3^{-1}$) leaching from soils results in the lower soil fertility, reduced crop productivity and increased water pollution. The effects of bentonite clay mixed with various nitrogen (N) fertilizers on $NO_3^{-1}$ leaching from sandy soils haven't been extensively studied. Therefore, the present lysimetric study determined $NO_3^{-1}$ leaching from bentonite [0, 2 and 4% (m/m)] treated sandy soil under three N sources (calcium nitrate [$Ca(NO_3)_2$], ammonium chloride [$NH_4Cl$], and urea [$CO(NH_2)_2$] at the rate of 300 kg N ha$^{-1}$). Results showed that bentonite markedly reduced $NO_3^{-1}$ release in the leachate, while 4% bentonite retained higher $NO_3$ in the soil. The $NO_3^{-1}$ leaching from sandy soil varied with N sources as $Ca(NO_3)_2$ > $NH_4Cl$ > $(CO(NH_2)_2$. At early stages of leaching, higher concentrations of $NO_3^{-1}$ were detected in leachate with both $NH_4Cl$ and $Ca(NO_3)_2$, but leaching of $NO_3^{-1}$ increased with urea at later leaching stages. The amount of total $NO_3^{-1}$ retained in soil was conversely related to the amount of $NO_3^{-1}$ in the leachate. This study indicated that soil amendment with bentonite could efficiently mitigate $NO_3^{-1}$ leaching from sandy soil and hence prevent N fertilizer losses and groundwater pollution.

## 1. Introduction

Growing populations and changing diets require an increase in agricultural production which may lead to an increase in the use of fertilizers. As such, food demand and fertilizer use have been forecasted to double or triple by 2050 [1]. Chemical N fertilizers and organic manures are often applied to the soil in higher amounts for higher agriculture production, which may lead to the N losses due to removal from the cropped fields into the water bodies [2] and/or emission into the atmosphere [3, 4]. Of the applied N for crops, only 40–50% is being incorporated into the agricultural products [5] and the remainder N is subjected to the substantial losses [6].

**Funding:** Yunnan Provincial Major Science and Technology Special Plan Projects (202102AE090030) Received by Tang Cheng.

**Competing interests:** The authors have declared that no competing interests exist.

The rainfall intensity and irrigation water influence the $NO_3^{-1}$ loss in the soil profile [7, 8]. The N fertilizers, used either as urea or ammonium form, are biochemically converted to $NO_3^{-1}$ which is susceptible to leaching from soil-plant system and enter groundwater bodies [2]. Therefore, an effective technology is required to prevent $NO_3^{-1}$ losses from sandy soils.

Soil $NO_3^{-1}$ is originated from both organic and inorganic N sources. Leaching and drainage studies found that $NO_3^{-1}$ is the major form of N occurred in the soil water [8–10]. A number of factors including plant characteristics, seasonal and climatic changes, and soil properties govern $NO_3^{-1}$ leaching from soils [11, 12]. The specific factors include soil texture, soil N concentration, amount of applied N, type of fertilizer, precipitation amount and intensity, soil water holding capacity, types of crops, root length and N demand of next crop [13, 14]. Leaching of $NO_3^{-1}$-N is more common than leaching of $NH_4$-N since both $NO_3^{-1}$-N and soil are negatively charged [15]. Over application or un-timely application of animal manures or commercial N fertilizers result in the nutrient imbalance in soils which lead to the increased N leaching rates, especially of $NO_3^{-1}$, into groundwater [12, 16]. Sandy soils, due to low water holding capacity [17], allow $NO_3^{-1}$ to leach down into the groundwater faster than the soils having fine textures, such as clay loams [13, 15]. Thus, leaching of $NO_3^{-1}$ through soil profile can potentially contaminate surface and groundwater [18]. Sandy soils with low organic matter may facilitate leaching of 10–15 mg $L^{-1}$ of $NO_3^{-1}$ to groundwater [19]. About 20–25% of this $NO_3^{-1}$ may enter surface water via buffer streams and wetlands causing eutrophication of water bodies [20, 21].

The increasing unsustainable agricultural use of N fertilizers results in $NO_3^{-1}$ leaching into ground waters [22, 23] and runoff into surface water ecosystem producing unfavorable consequences [24], which adversely affect water quality [22, 25]. The increasing potential of contamination of water resources is linked with the inefficient management of N fertilizer when compared with the natural systems [26–28]. The concentration of $NO_3^{-1}$ above 10 mg $L^{-1}$ in drinking water are considered as harmful for human health [29]. Higher $NO_3^{-1}$ consumption has been affiliated with various illnesses, e.g., methemoglobinemia has been proven due to ingestion of over nitrate concentrations in water [30, 31]. The endogenous $NO_3^{-1}$ may chemically be transformed to carcinogenic N compounds leading to adverse effects of colorectal cancer [32] and bladder cancer [33]. Therefore, developing an effective technology to retain nutrients in soils is imperative to prevent $NO_3^{-1}$ leaching from soils. Soil amendments have been considered as management practice to reduce $NO_3^{-1}$ losses from sandy soils [34]. Bentonite, an alumina-siliceous clay material, has not been previously utilized to control $NO_3^{-1}$ leaching.

Bentonite, like other clays, are hydrous aluminosilicates with fine colloids of < 2 mm of soils [35]. Clays are composed of fine-grained clays minerals and crystals such as quartz, carbonates and oxides [35] and are considered to retain contaminants by anion and cation exchange processes and prevent leaching into groundwaters. Due to effective adsorption capacities, bentonite clay has been used for multiple purposes. Bentonite is also used to remove dyes, radioactive waste, purification of viral RNA and wastewater [36–38]. Bentonite application as amendment enhanced soil fertility by increased soil carbon and potassium [39], while improved water holding capacity of sandy soils under drought stress [40]. Bentonite application to sandy acidic soil improved soil fertility by increasing availability of macro-nutrient (up to 30%) to plants [41]. Fertilizers, if used in combination with nano-dimensional adsorbents increase nutrient use efficiency and reduce nutrient leaching into groundwaters [42], Clay amended sandy soil significantly reduced N and P leaching by 20% to 60% [43]. Leaching of $NH_4$-N was reduced by 70% from a mixture of biochar, urea and bentonite plus sepiolite clay [44]. However, it is still unclear that how the type of fertilizer and application of bentonite clay to soils can mitigate $NO_3^{-1}$ leaching.

A reduction in the $NO_3^{-1}$ leaching was expected when clay material was applied to the soil. Reports evaluating the interactive effects of bentonite material and N sources on the reduction of $NO_3^{-1}$ leaching from sandy soils are scanty. Therefore, the objective of the present study was to investigate the influence of bentonite on $NO_3^{-1}$ leachability from a sandy soil after application of calcium nitrate [$Ca(NO_3)_2$], ammonium chloride [$NH_4Cl$], and urea [$CO(NH_2)_2$] as three N sources.

## 2. Materials and methods

### 2.1 Lysimeter experiment

A leaching experiment was conducted in the Soil Science Laboratory at COMSATS University Islamabad, Abbottabad Campus, Pakistan following the idea of Zhao et al. [45]. For this purpose, PVC columns, with 0.60 m length and 0.15 m diameter, were installed to run the experiment. The bottles used for collecting leachate were installed on the floor. Filter papers were placed on porous bottom of the columns to prevent soil leaching. The columns were connected with bottles using small pipes for the collection of leachate. The connecting pipes were kept airtight to prevent evaporation from the leachate bottles. Locally collected sandy soil samples from agricultural land (0–12 cm depth) were utilized for the experiment. Bentonite material was commercially purchased and then air-dried. Both the soil and bentonite were analyzed for physico-chemical properties before the experiment. After air drying, 46 kg of the soil was added to each column (0.50 m length). The air-dried clay was applied at the rate of 0, 2 and 4% to the sandy soil packed in a PVC column. Three treatments of nitrogen (N) fertilizers namely calcium nitrate [$Ca(NO_3)_2$], ammonium chloride [$NH_4Cl$], and urea [$CO(NH_2)_2$] were applied to the soil. Based on the bulk density of soil (1.3 g cm$^{-1}$), each fertilizer was applied at 300 kg N ha$^{-1}$. Initially, as the soil was dry, the amount of first water application was kept higher so that enough water may drain out to collect the leachate. For later leaching events, the leaching fraction (LF) was calculated by dividing the drained water by applied water. Then, the tap water was applied at leaching fraction of 0.3~0.4. Leachate was collected within 24 h after each water application. A total of five leaching events were covered, that is, 1st, 2nd, 4th, 6th and 10th day. The graphical display of the experiment is illustrated in [Fig 1].

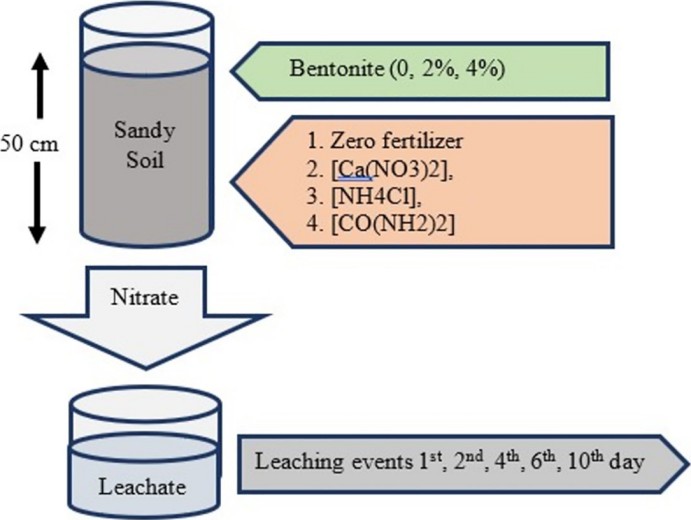

**Fig 1. Graphical display of lysimeter experiment.**

## 2.2 Laboratory analysis

Chemical analyses of soil, bentonite and tap water were carried out before experiment. Soil samples were air-dried and sieved via 2 mm sieve. The soil and air-dried bentonite material were tested for pH and electrical conductivity (EC) in 1:5 (w/v) soil-water suspensions by a pH meter (Model: HANNA HI 8520) and EC meter (Model: 4320 JENWAY), respectively [46]. The soil was saturated for overnight, weighed and then the water holding capacity (WHC) of the soil was calculated by the difference in the weight of soil [47]. The bulk density of soil was determined using cylindrical cores. The soil sample was weighed and placed in the oven at 105˚C for 8 to 12 hours until the weight was constant. Bulk density was then calculated in the same way as described by Grossman and Reinsch [48].

The post-experiment soil was sampled in two layers (0–25 and 25–50 cm) from the PVC columns and was thoroughly mixed. A 10 g soil sample was shaken in 100 mL distilled water for 1 h and then the suspension was filtered. Moisture content in the soil samples was adjusted by oven drying few grams of soil. The leachate collected from 5 events was analyzed for $NO_3$ concentration. The $NO_3^{-1}$ concentration in pre- and post-experiment soil, bentonite and leachate was determined by UV spectrophotometer (Model: LI-UV-7000) at 220 nm [49]. All the reagents/chemicals of Sigma Aldrich, Germany, were utilized during the experiment.

## 2.3 Statistical analysis

Data were statistically analyzed by OriginLab 2021 for graphical presentation. The three-way analysis of variance (ANOVA) was performed on Sigmaplot. The three factors were taken as bentonite (0, 2% and 4%), N fertilizer sources ($[Ca(NO_3)_2]$, $[NH_4Cl]$ and $[CO(NH_2)_2]$), and leaching events (1st, 2nd, 4th, 6th and 10th day) with three replications. A post hoc Tukey test was also performed to determine the significant difference between the levels of factors.

# 3. Results

## 3.1. Pre-experiment chemical analysis

Bentonite clay, soil and water were analyzed for chemical properties before experiment, which are presented in Table 1. Analysis revealed that soil had highest $NO_3^{-1}$ concentrations, compared to bentonite clay. The tap water had low concentrations of $NO_3^{-1}$. The electrical conductivity (EC) of bentonite was highest compared to soil and water samples, but still it fell below the category of non-saline (EC<4 dS $m^{-1}$). However, the pH of bentonite was lower (pH<7) making it more acidic as compared to soil and water.

## 3.2. Effect of bentonite and N sources on $NO_3^{-1}$ leaching

Impact of bentonite clay mixed with different N sources on $NO_3^{-1}$ leaching is illustrated in [Fig 2]. Results showed $NO_3^{-1}$ concentration decreased by 12 to 19% in the leachate, irrespective of the source of N with increasing bentonite rates. The significantly highest reduction (20–25%) in $NO_3^{-1}$ leaching was recorded with $Ca(NO_3)_2$ with 4% bentonite as compared with $CO(NH_2)_2$ and $NH_4Cl$ at similar bentonite rates. At early stages of leaching, the leachate showed

**Table 1. Nitrate, electrical conductivity (EC) and pH of bentonite clay material, soil and tap water.**

| Material | Nitrate | EC (mS $m^{-1}$) | pH |
|---|---|---|---|
| Bentonite | 23.5 mg $kg^{-1}$ | 126.7 | 5.7 |
| Soil | 34.4 mg $kg^{-1}$ | 87.8 | 7.8 |
| Tap water | 7.8 mg $L^{-1}$ | 37.5 | 7.2 |

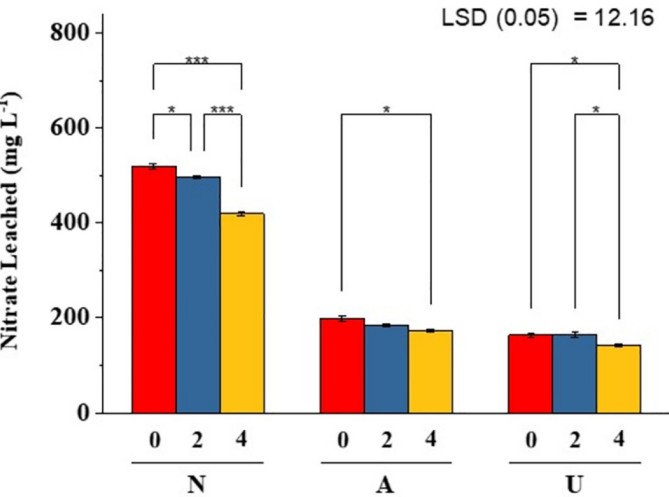

**Fig 2. Average amount of leachate nitrate from bentonite treated sandy soil (from day 1 to day 10).** N, A and U indicate $Ca(NO_3)_2$, $NH_4Cl$ and $CO(NH_2)_2$, respectively. Bentonite clay was applied at the rate of 0%, 2% and 4%. *, ** and *** show significant differences at P<0.05, P<0,01 and P<0.001, respectively.

higher concentrations of $NO_3^{-1}$ in leachate with both $NH_4Cl$ and $Ca(NO_3)_2$, but leaching of $NO_3$ increased with urea sources at later leaching stages [Fig 3]. Total $NO_3^{-1}$ loads were higher in soil with urea and $Ca(NO_3)_2$ treated soil at 4% bentonite as compared to $NH_4Cl$ [Fig 4]. The incubation of soil with bentonite (4%) reduced $NO_3$ content by 7%, 20% and 8% with $Ca(NO_3)_2$, $NH_4Cl$ and $CO(NH_2)_2$ treated soil, respectively [Fig 5].

### 3.3. Statistical analysis

A three-way analysis of variance (ANOVA) was performed on $NO_3^{-1}$ leaching with 3 N sources, 3 levels of bentonite and 5 leaching events [Table 2]. Analysis showed that there was significant difference (P<0.001) within N sources, bentonite levels and leaching events (days). There was statistically significant (P<0.01) interaction among all the factors. The Tukey test revealed significant (P<0.05) difference between 0% and 2%, and 0% and 4% bentonite with all N sources on $NO_3^{-1}$ leaching, but the difference was not significant between 2% and 4% bentonite with all N sources [Table 2].

## 4. Discussion

Results showed that irrespective of the source of N, the $NO_3^{-1}$ leaching consistently decreased with increasing bentonite application showing the sequences of fertilizer type as: $Ca(NO_3)_2$ > $NH_4Cl$ > $CO(NH_2)_2$ [Fig 2]. Form of N leached from the soil columns was closely related to the type of fertilizer applied to the soil [50]. The $NO_3^{-1}$ fertilizers appeared to be more sensitive to the leaching, especially in sandy soils [51] and also to the denitrification [52] as compared to urea or ammonium fertilizers. Also, $NO_3^{-1}$ leaching is substantially higher in free-drained soils [53], such as sandy soil with macropores used in the present study.

Application of bentonite decreased $NO_3^{-1}$ leaching, regardless to the rate of application. The $[NO_3^{-1}]$ decreased with increasing bentonite treatments level. Whereby higher $NO_3^{-1}$ concentration was observed in $NO_3^{-1}$ and $NH_4$ containing fertilizers during the initial leaching [Fig 3]. Urea form of N showed consistent increases in $NO_3^{-1}$ concentrations in water collected in later leaching stages [Fig 3]. This could be associated with increased nitrification process in

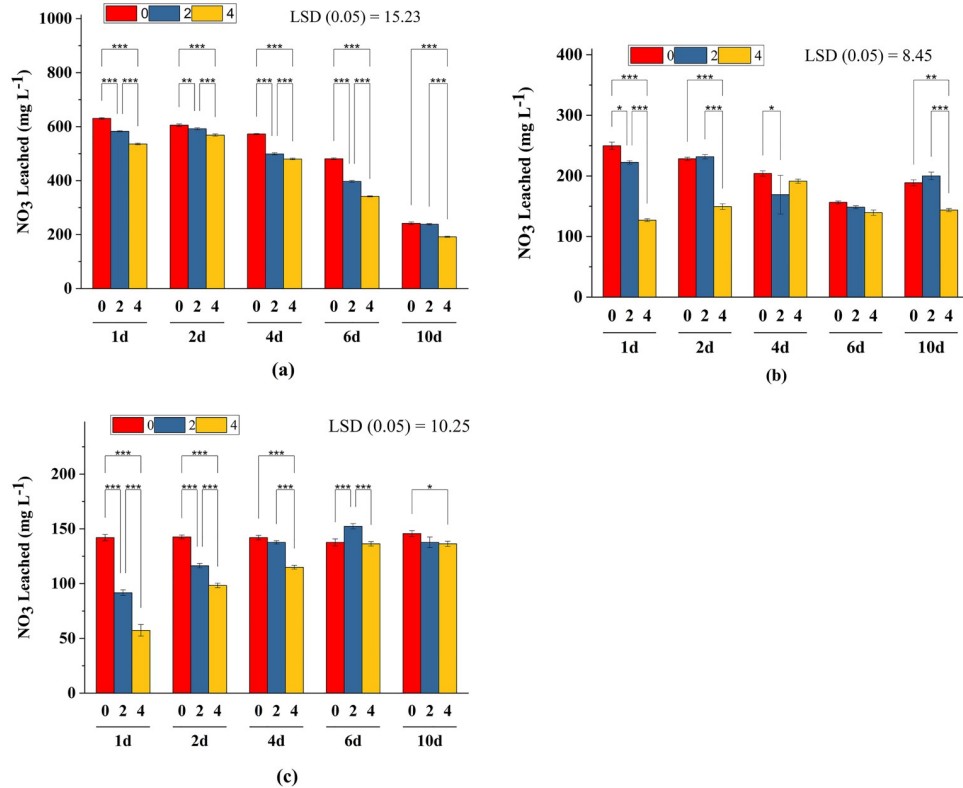

**Fig 3. Nitrate leaching in bentonite amended sandy soil using $Ca(NO_3)_2$ (a), $NH_4Cl$ (b), and $CO(NH_2)_2$ (c) with 0%, 2% and 4% bentonite during five leaching events (days).** *, ** and *** show significant difference at P<0.05, P<0,01 and P<0.001, respectively.

soils under unsaturated conditions [54], which might have resulted in increased $NO_3^{-1}$ leaching at later stages. Accumulation of $NO_3^{-1}$ was more in the soil sampled from the lower layer of the column after a leaching process, showing the sequence as $CO(NH_2)_2$ > $NH_4Cl$ > $Ca(NO_3)_2$.

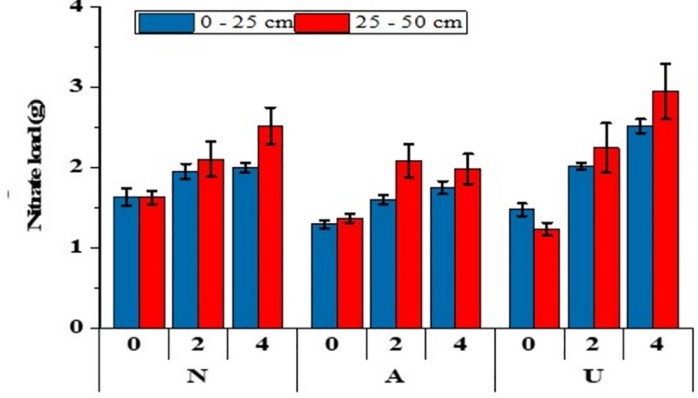

**Fig 4. Residual nitrate loads (g) in soil after leaching events (days) at 0%, 2% and 4% bentonite with $Ca(NO_3)_2$ (N), $NH_4Cl$ (A), and $CO(NH_2)_2$ (U).** *, ** and *** show significant difference at P<0.05, P<0,01 and P<0.001, respectively.

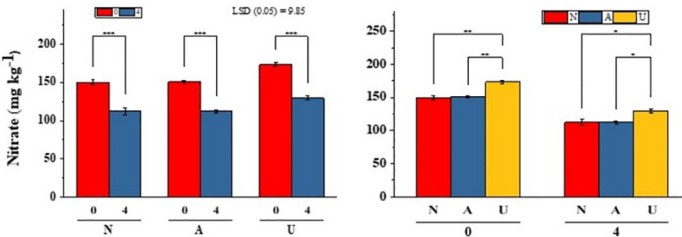

**Fig 5. Nitrate concentration in 4% bentonite clay mixed sandy soil after discrete incubation with $Ca(NO_3)_2$ (N), $NH_4Cl$ (A) and $CO(NH_2)_2$ (U), *, ** and *** show significant difference between treatments at P<0.05, P<0,01 and P<0.001, respectively.**

The amount of total $NO_3^{-1}$ retained in soil was termed as nitrate loads conversely related to the amount of $NO_3^{-1}$ in the leachates [Fig 4]. An enhanced application of bentonite significantly retained $NO_3^{-1}$ in the soil columns. A higher amount of $NO_3^{-1}$ was retained in the soil amended with 4% bentonite. The application of bentonite clay enhanced soil moisture and improved macro-aggregate development [55] which improved soil quality through structural development, by increased exchange of anions and cations [56] and helped in reduced leaching while promoting nutrient retention [57].

Enrichment of sandy soils with bentonite increased the porosity and altered the pore-size distribution [58]. The interactions of bentonite with biochar and urea improved soil properties by diffusing soil moisture which controlled the mobility of nutrients within soils [59], thus with high water retaining capacity, increased exchange capacity, swelling, thermal stability and slow-releasing characteristics, bentonite offers valuable solution to reduced nutrient leaching from loose soils.

Higher quantity of $NO_3^{-1}$ was retained in urea treated soil followed by nitrate and ammonium containing fertilizers [Fig 4]. Such retention could be attributed to the transformation of $NO_3^{-1}$ in urea contained soil after few days of incubation. Across all N sources, the application of bentonite (4%) markedly limited the release of $NO_3^{-1}$. After incubation, bentonite contents reduced the magnitude of $NO_3^{-1}$ among fertilizers as follow: 7% in $Ca(NO_3)_2$, 20% in $NH_4Cl$ and 8% in urea treated soil [Fig 5]. The soil having negatively charged sites attracts more positively charged $NH_4$ as compared to the negatively charged $NO_3^{-1}$, and therefore the $NH_4$ has been considered to be a less mobile in soils, than $NO_3^{-1}$ [60, 61].

The effect of bentonite on the leachability of $NO_3^{-1}$ varied with different N fertilizers. The $NO_3^{-1}$ leaching consistently decreased with increasing bentonite application showing the

**Table 2. Summary ANOVA on effect of bentonite and N sources on NO3-1 leaching at different leaching events.**

| Source of Variation | DF | SS | MS | F | P |
|---|---|---|---|---|---|
| N sources (N) | 2 | 2943336.13 | 1471668 | 14453 | < 0.01 |
| Bentonite levels (B) | 2 | 74157.91 | 37078 | 364 | < 0.01 |
| Leaching event / day (D) | 4 | 282061.82 | 70515 | 692 | < 0.01 |
| N x B | 4 | 11270.48 | 2817 | 27 | < 0.01 |
| N x D | 8 | 589355.42 | 73669 | 723 | < 0.01 |
| B x D | 8 | 15955 | 1994 | 19 | < 0.01 |
| N x B x D | 16 | 24125 | 1507 | 14 | < 0.01 |
| Residual | 90 | 9164 | 101 | | |
| Total | 134 | 3949426 | 29473 | | |

sequences of fertilizer type as: $Ca(NO_3)_2 > NH_4Cl > CO(NH_2)_2$ [Fig 2]. Addition of Ca as Ca$(NO_3)_2$ increased the adsorptive capacity of bentonite at low pH (5–6), while higher concentration of $NO_3^{-1}$ ion due to the addition of calcium nitrate as N source resulted in increased adsorption of $NO_3^{-1}$ at bentonite clay surfaces [62]. The mechanism can be further explained by Ca hydrolysis, which resulted into the formation of less soluble $Ca(OH)_2$, releasing more $H^+$ ions, thus acidifying the media [Eq 1] [63]. Under low pH, the anion exchange capacity of bentonites in significantly increased which offers more positive sites to attract $NO_3^{-1}$ on its surface [Eq 2], which increased the retention of $NO_3^{-1}$ ions on soil colloids due to adsorption phenomenon [Eq 3]. This ultimately reduced $NO_3^{-1}$ leaching from sandy soil and showed lower $NO_3^{-1}$ concentrations in leachate. The entire process can be explained as follows, where A is dissociated anion and X is the soil colloidal surface

$$Ca^{2+} + H_2O \rightleftharpoons Ca(OH)_2 + 2H^+ \tag{1}$$

$$X + H^+ \rightleftharpoons X^+ + HA \tag{2}$$

$$X^+ + NO_3^- \rightleftharpoons XNO_3 \tag{3}$$

Surface charge of variable charge clays varies with changes in pH of soil solution, therefore, at low pH, the anion exchange capacity exceeds cation exchange capacities which retain more $NO_3^{-1}$ on its surfaces [64], while increasing the mobility of $NH_4$ in such conditions. However, the mobility of both N forms can be maintained by adjusting the rates of $Ca(NO_3)_2$ and urea along with bentonite clay under field conditions. The study suggested that bentonite amendment with $Ca(NO_3)_2$ as N sources could effectively mitigate $NO_3^{-1}$ leaching from sandy soils.

## 5. Conclusions

It is concluded that N sources and bentonite application were important factors affecting the $NO_3^{-1}$ leaching from sandy soil. Application of N sources enhanced $NO_3^{-1}$ leaching from the sandy soil. The $NO_3^{-1}$ leaching decreased in the order of $Ca(NO_3)_2 > NH_4Cl >$ urea. Bentonite substantially reduced $NO_3^{-1}$ in the leachate. Urea showed higher $NO_3^{-1}$ at the later leaching events. Higher contents of $NO_3^{-1}$ were retained in the soil with 4% bentonite. Higher $NO_3^{-1}$ contents were accumulated in the lower part of the soil column after a leaching process. This experiment suggests that bentonite clay chemistry with added $Ca(NO_3)_2$ provide better understanding of anion exchange capacity to retain higher $NO_3^{-1}$ concentrations in soil, thereby decreasing $NO_3^{-1}$ leaching from sandy soil. Further research is suggested to investigate the effects of different clay types on the dynamics of nitrate under field conditions.

## Supporting information

**S1 File.**
(PDF)

## Author Contributions

**Conceptualization:** Muhammad Irshad.

**Data curation:** Tang Cheng.

**Formal analysis:** Muhammad Mohiuddin.

**Funding acquisition:** Tang Cheng.

**Software:** Chen Yao.

**Supervision:** Muhammad Irshad.

**Visualization:** Di Song.

**Writing – original draft:** Zahid Hussain.

**Writing – review & editing:** Riaz Ahmed Khattak.

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
