## [Decision Letter · Decision Letter 0]

5 Aug 2022

PONE-D-22-14894Bentonite clay mixed with different N sources significantly reduce nitrate leaching from sandy soilPLOS ONE

Dear Dr. Zahid Hussain,

Thank you for submitting your manuscript to PLOS ONE. After careful consideration, we feel that it has merit but does not fully meet PLOS ONE’s publication criteria as it currently stands. Therefore, we invite you to submit a revised version of the manuscript that addresses the points raised during the review process.

The title and introduction of this manuscript needs to be revised thoroughly. Similarly, there are some serious concerns existed on English language used in this manuscript. 

We look forward to receiving your revised manuscript.

Kind regards,

Waqas ud Din Khan, Ph. D.

Academic Editor

PLOS ONE

Journal Requirements:

"The authors received no specific funding for this work"

"Authors appreciate the support of Yunnan Provincial Key Laboratory of Pollution Processes and Management in Plateau Lake Watershed (Project No: 202005AG070105) for funding this publication."

"The authors received no specific funding for this work"

Reviewers' comments:

Reviewer's Responses to Questions

**Comments to the Author**

1. Is the manuscript technically sound, and do the data support the conclusions?

Reviewer #1: Yes

Reviewer #2: Partly

Reviewer #3: Yes

2. Has the statistical analysis been performed appropriately and rigorously? 

Reviewer #1: Yes

Reviewer #2: Yes

Reviewer #3: Yes

3. Have the authors made all data underlying the findings in their manuscript fully available?

Reviewer #1: Yes

Reviewer #2: Yes

Reviewer #3: Yes

4. Is the manuscript presented in an intelligible fashion and written in standard English?

Reviewer #1: Yes

Reviewer #2: Yes

Reviewer #3: Yes

5. Review Comments to the Author

Reviewer #1: Reviewed manuscript “Bentonite clay mixed with different N sources significantly reduce1

nitrate leaching from sandy soil” is an original and interesting study. Authors comprehensively evaluated the nitrate leaching in sandy soil that is beneficial for both environment and agriculture. I would suggest minor revision.

Following are some suggestions for further improvements:

From the title, I would suggest you to remove the words “mixed” and “significantly” to make it more catchy.

In abstract results should be extended to few more lines.

Journal format need to be followed regarding citations write up in text.

Introduction and Discussion section needs to further strengthen by latest studies on the subject.

Line 105: Company name and details for used material should incorporate

Line 106: for chemical for physical and chemical properties should be corrected

Line 120: Sentence not correct

Figure legends are too brief.

At some places in the text, there are grammatical mistakes that needs to be corrected by some native English colleague.

To further strengthen introduction, following latest studies etc. are suggested to cite:

https://doi.org/10.1016/j.plantsci.2019.110270

https://doi.org/10.1080/01904167.2019.1648669

Reviewer #2: General remarks:

1- The manuscript has an average quality of data. There must be some more experiments which supports different adsorption models to make the quality of this manuscript more better

2- The introduction part has not enough citations which is necessary to support your statements of manuscript.

3- Most of the references are old, there must be recent references from 2018 to onwards.

4- In the whole manuscript please replace NO3 to NO3−1

5- The discussion has been elaborated in a good way but it needs recent references.

6- Format all the references in the reference list according to PLOS ONE journal. https://www.nlm.nih.gov/bsd/uniform_requirements.html

Comment 1: please edit the title from “Bentonite clay mixed with different N sources significantly reduce nitrate leaching from sandy soil” to “Bentonite clay mixed with different nitrogen sources, significantly reduce nitrate leaching from sandy soil.”

Comment 2: line number 44 should be like “may lead to an increase in the use of fertilizers.” Instead of “will/may lead to an increase in the use of fertilizers.”

Comment 3: in the start of line 47 there is no reference cited after the word atmosphere.

Comment 4: from line 50 to 53 “Developing an effective technology to retain nutrients in soils is imperative to prevent NO3 leaching from soils. Soil amendments have been considered as management practice to reduce NO3 losses from sandy soils (Irshad et al., 2014). Bentonite, an alumina-siliceous clay material, has not been previously utilized to control NO3 leaching.”

There is no synchronization with the ongoing paragraph.

Comment 5: from line 47 to line 51 there is no single reference has been observed. Please support your statements with published research work.

Comment 6: in line 59 there must be root length instead of rooting length.

Comment 7: line 77 to 78 there is no supporting reference has been found. Please write a proper reference.

Comment 8: from line 84 to 87, the author discussed about different amendments to control nitrogen leaching. Author must elaborate that how many percent the amendment improved the nitrogen leaching as compare to control treatment.

Comment 9: in the last paragraph of introduction from 92 to 96, there must be proper objectives of the aimed study. Please write it.

Comment 10: in line 97 “Lysimeter experiment” has been described well but there is no supporting reference which describe that from where the author generated the idea.

Comment 11: in Fig.1, author must use subscript letters for representing the chemical formulas e.g [CO(NH2)2] should be written as [CO(NH2)2] please do it for the rest also.

Comment 12: in the line 123, the author discussed about water holding capacity analysis but did not mentioned the reference at the end. Please cite a reference to support the analysis method.

Comment 13: why there are double brackets in line 137 and 138 “((0 fertilizer, [Ca(NO3)2], [NH4Cl] and [CO(NH2)2])).” Please remove them.

Comment 14: from line 242 to 249 there is no supporting reference has been cited.

Comment 15: in the line 258, the NO3 should be written as NO3.

Reviewer #3: I would suggest to authors to address following comments,

How different sources of N applied NO3 leaching from sandy soil? What could be mechanistic explanation of this process.

Line 45-47, should be supported by doi.org/10.1007/s11356-016-7894-4

Significant amount of references are more than 15-20 years old, they need to be updated. For example, The United States Environmental Protection Agency set the standard for drinking water and health advisory level of 10 mg L-1 nitrate based on the human health risks (Kross et al., 1993), which is a very old figure.

Authors did not develop a link between NO3 leaching and sandy soil, they have provided a generalized overview of NO3 leaching in soil, but did not explain in the context of sandy soil and why authors have used sandy soil in this study?

6. PLOS authors have the option to publish the peer review history of their article (what does this mean?). If published, this will include your full peer review and any attached files.

Reviewer #1: No

Reviewer #2: **Yes: **Fiza Pir Dad

Reviewer #3: No

---

## [Author Response · Author response to Decision Letter 0]

30 Sep 2022

A detailed "Response to Reviewers" file has been uploaded. 

Two reviewers (1 and 2) suggested changes in Title of the manuscript. The title was modified keeping in view of both reviewers. However, it can be further modified if suggested

New references has been added as suggested by all reviewers

---

## [Decision Letter · Decision Letter 1]

4 Nov 2022

PONE-D-22-14894R1Bentonite clay with different nitrogen sources reduce nitrate leaching from sandy soilPLOS ONE

Dear Dr. Zhaid Hussain

Thank you for submitting your manuscript to PLOS ONE. After careful consideration, we feel that it has merit but does not fully meet PLOS ONE’s publication criteria as it currently stands. Therefore, we invite you to submit a revised version of the manuscript that addresses the points raised during the review process.

We look forward to receiving your revised manuscript.

Kind regards,

Waqas ud Din Khan, Ph. D.

Academic Editor

PLOS ONE

Journal Requirements:

Reviewers' comments:

Reviewer's Responses to Questions

**Comments to the Author**

1. If the authors have adequately addressed your comments raised in a previous round of review and you feel that this manuscript is now acceptable for publication, you may indicate that here to bypass the “Comments to the Author” section, enter your conflict of interest statement in the “Confidential to Editor” section, and submit your "Accept" recommendation.

Reviewer #2: All comments have been addressed

Reviewer #3: All comments have been addressed

2. Is the manuscript technically sound, and do the data support the conclusions?

Reviewer #2: Partly

Reviewer #3: Yes

3. Has the statistical analysis been performed appropriately and rigorously? 

Reviewer #2: Yes

Reviewer #3: Yes

4. Have the authors made all data underlying the findings in their manuscript fully available?

Reviewer #2: Yes

Reviewer #3: Yes

5. Is the manuscript presented in an intelligible fashion and written in standard English?

Reviewer #2: Yes

Reviewer #3: Yes

6. Review Comments to the Author

Reviewer #2: line no 105: please write here three sources names in objectives.

Line 32: The effects of bentonite clay on NO3 -1leaching from sandy soils are not

extensively studied. Author is sure that he is not using the sandy soil?

Line 55: The N fertilizers, used either as urea or ammonium form, is biochemically converted to NO3-1 which is susceptible to leaching from soil-plant system and enter groundwater bodies.

Please rewrite the line by replacing is with are.

Line 61, 73: NO3 should be replaced with NO3-1.

Reviewer #3: Authors have revised the manuscript and can be accepted in current form for publication. This is a very interested study and have showed the combination of Bentonite clay with different nitrogen sources to reduce N leaching from sandy soils.

7. PLOS authors have the option to publish the peer review history of their article (what does this mean?). If published, this will include your full peer review and any attached files.

Reviewer #2: **Yes: **Fiza Pir Dad

Reviewer #3: No

---

## [Author Response · Author response to Decision Letter 1]

7 Nov 2022

A separate file "Response to Reviewers" has been uploaded. It includes detailed response to the comments by reviewers.

---

## [Decision Letter · Decision Letter 2]

24 Nov 2022

Bentonite clay with different nitrogen sources can effectively reduce nitrate leaching from sandy soil

PONE-D-22-14894R2

Dear Dr. Zahid Hussain,

We’re pleased to inform you that your manuscript has been judged scientifically suitable for publication and will be formally accepted for publication once it meets all outstanding technical requirements.

Kind regards,

Waqas ud Din Khan, Ph. D.

Academic Editor

PLOS ONE

Additional Editor Comments (optional):

Reviewers' comments:

Reviewer's Responses to Questions

**Comments to the Author**

1. If the authors have adequately addressed your comments raised in a previous round of review and you feel that this manuscript is now acceptable for publication, you may indicate that here to bypass the “Comments to the Author” section, enter your conflict of interest statement in the “Confidential to Editor” section, and submit your "Accept" recommendation.

Reviewer #2: All comments have been addressed

2. Is the manuscript technically sound, and do the data support the conclusions?

Reviewer #2: Partly

3. Has the statistical analysis been performed appropriately and rigorously? 

Reviewer #2: Yes

4. Have the authors made all data underlying the findings in their manuscript fully available?

Reviewer #2: Yes

5. Is the manuscript presented in an intelligible fashion and written in standard English?

Reviewer #2: Yes

6. Review Comments to the Author

Reviewer #2: The manuscript is in good from now. Author has successfully addressed all the comments according to my expertise. I recommend this article to publish in Plos One.

7. PLOS authors have the option to publish the peer review history of their article (what does this mean?). If published, this will include your full peer review and any attached files.

Reviewer #2: **Yes: **Fiza Pir Dad

---

## [Editor Report · Acceptance letter]

9 Dec 2022

PONE-D-22-14894R2 

Bentonite clay with different nitrogen sources can effectively reduce nitrate leaching from sandy soil 

Dear Dr. Hussain:

I'm pleased to inform you that your manuscript has been deemed suitable for publication in PLOS ONE. Congratulations! Your manuscript is now with our production department. 

Kind regards, 

on behalf of

Dr. Waqas ud Din Khan 

Academic Editor

PLOS ONE